# Identification of Ku70 Domain-Specific Interactors Using BioID2

**DOI:** 10.3390/cells10030646

**Published:** 2021-03-14

**Authors:** Sanna Abbasi, Caroline Schild-Poulter

**Affiliations:** Robarts Research Institute and Department of Biochemistry, Schulich School of Medicine and Dentistry, University of Western Ontario, London, ON N6A 5B7, Canada; sabbasi5@uwo.ca

**Keywords:** BioID2, proximity-dependent biotin identification, Ku70, von Willebrand A-like domain, RNF113A, Spindly

## Abstract

Since its inception, proximity-dependent biotin identification (BioID), an in vivo biochemical screening method to identify proximal protein interactors, has seen extensive developments. Improvements and variants of the original BioID technique are being reported regularly, each expanding upon the existing potential of the original technique. While this is advancing our capabilities to study protein interactions under different contexts, we have yet to explore the full potential of the existing BioID variants already at our disposal. Here, we used BioID2 in an innovative manner to identify and map domain-specific protein interactions for the human Ku70 protein. Four HEK293 cell lines were created, each stably expressing various BioID2-tagged Ku70 segments designed to collectively identify factors that interact with different regions of Ku70. Historically, although many interactions have been mapped to the C-terminus of the Ku70 protein, few have been mapped to the N-terminal von Willebrand A-like domain, a canonical protein-binding domain ideally situated as a site for protein interaction. Using this segmented approach, we were able to identify domain-specific interactors as well as evaluate advantages and drawbacks of the BioID2 technique. Our study identifies several potential new Ku70 interactors and validates RNF113A and Spindly as proteins that contact or co-localize with Ku in a Ku70 vWA domain-specific manner.

## 1. Introduction

Proximity-dependent biotin identification (BioID) was first published in 2012 as a novel biochemical technique for identifying proximal protein interactors in vivo [1]. This technique relies on fusing a bait protein of interest with a mutant promiscuous biotin ligase enzyme capable of in vivo tagging or “biotinylating” nearby proteins with a biotin molecule. Following the biotin incubation period, biotinylated proteins can be isolated using streptavidin-conjugated resins and then identified by mass spectrometry. BioID has many advantages over traditional protein interaction identification methods such as co-immunoprecipitation or yeast-two hybrid experiments. One advantage is the employment of the biotin-streptavidin bond, one of the strongest non-covalent bonds in nature, allowing protein isolation to occur under stringent conditions. A second advantage is that this technique can capture associations that occur in vivo, sometimes providing enough information to depict the context for interaction. This technique allows for the identification of proteins that interact directly, indirectly, or are within proximity (~10 nm) to the bait protein [1].

The BioID method originally used a biotin ligase enzyme isolated from *Escherichia coli*, however an improvement on the technique called BioID2, published in 2016, used the biotin ligase isolated from *Aquifex aeolicus* [2]. Compared to its predecessor, BioID2 offers improvements in terms of more efficient biotin tagging (biotinylation) and greater sensitivity to nearby proteins [2]. Since then, others have continued to build on the momentum of this technique and have developed novel variations of the original method, many of which have been published recently (reviewed in [3]). Since its inception, BioID has provided the basis for various other techniques such as in vivo BioID (iBioID) [4], split-BioID [5], RaPID [6], TurboID/miniTurbo [7], 2C-BioID [8], in vitro BioID (ivBioID) [9], and split-TurboID [10]. While this rapid development is expanding our toolbox and providing more specialized techniques to identify protein interactions under different contexts, we have yet to fully explore the potential of the existing BioID variants already at our disposal. Very recently, efforts have been made to explore the greater accessibility of intrinsically disordered regions using the BioID method [11] and to compare different proximity-based methods, BioID versus TurboID [12]. These types of studies are particularly valuable as they advance our understanding of the limitations of the BioID technique. For example, while one of the greatest advantages of BioID is that it can theoretically tag and identify all protein interactors in vivo (including weak, transient, and proximal interactions), the validation of candidate interactors can prove difficult using gold-standard methods such as co-immunoprecipitation or in vitro binding assays, which detect strong protein interactions. In addition, the pipeline for how best to filter BioID candidates to identify true positives while removing false positives is still a process in development. Notably, many algorithms (MiST [13], SAINT [14], SAINTexpress [15], MUSE [16]) have been developed which rely on quantitative analysis to filter candidates.

While both BioID and BioID2 can identify protein interactors, their capability in mapping those identified candidates to specific protein domains or regions within the bait protein has yet to be fully explored. The concept of using protein domains as bait has previously been employed to successfully identify protein interactors with methods such as co-immunoprecipitation assays, yeast two-hybrid studies, and quantitative fragmentome mapping [17]. As an example related to this study, Grundy et al. (2016) used the von Willebrand A (vWA)-like domain from the Ku80 (*XRCC5*) protein in a yeast two-hybrid screen and were able to identify the Ku80 vWA-specific interactor, CYREN/MRI [18].

Generally, large proteins can be more challenging to study biochemically. This is exacerbated with the sizable addition of the biotin ligase needed for BioID and BioID2, representing another limitation of the method. Using protein domains as the bait, rather than the full-length protein, could expand upon the potential for BioID/BioID2 to identify protein interactors for larger proteins.

BioID and BioID2 have thus far mostly been conducted with full-length proteins, with only a few studies attempting to use non-full-length bait proteins (i.e., bait proteins with deletions) [19,20]. We have previously used BioID to identify proteins interacting with human Ku70 [21]. Ku is a highly abundant protein complex in humans, composed of subunits Ku70 and Ku80, that are believed to exist as an obligate heterodimer [22]. Ku subunits closely intertwine to form a basket-shaped heterodimer capable of encircling double-stranded DNA ends [22]. Ku is best characterized for its role in repairing double-stranded breaks (DSBs) through the DNA repair pathway non-homologous end-joining (NHEJ), although it has been implicated in many other cellular processes [23]. Collectively, Ku has been found to interact with many proteins [24,25,26,27,28,29,30,31,32,33,34,35,36,37,38,39,40,41,42,43,44,45,46,47,48,49,50,51,52,53,54,55,56], with some interactions being mapped specifically to the Ku80 subunit [18,57,58,59,60,61] and others to the Ku70 subunit [54,57,62,63,64,65,66,67,68,69,70,71,72,73,74,75,76,77,78,79,80,81,82,83,84], though notably few interactors were mapped to the Ku70 vWA domain [85,86,87,88] (Figure 1). Here, we used Ku70 to conduct BioID2, but this time we expanded upon the original technique to assess its ability to identify and map domain-specific interactors. To do this, we conducted BioID2 with the Ku70 vWA domain deleted and with the Ku70 vWA domain alone.

We chose to study Ku70 for the following reasons: first, our lab had already conducted BioID with full-length Ku70, thus we had some guidance for our experimental design. Second, there are already established, direct protein interactors that have been mapped to specific regions of Ku70 (Figure 1), allowing us to validate the approach. Third, because full-length Ku70 normally forms a heterodimer with Ku80, using a specific Ku70 domain incapable of forming the heterodimer would allow us to exclude Ku80 interactors. Fourth, since only a few proteins have been found to interact with the Ku70 vWA domain, we were interested in identifying novel interactors for this specific region.

## 2. Materials and Methods

### 2.1. Designing and Cloning Plasmid Expression Constructs

The wild-type human Ku70 (*XRCC6*) cDNA was PCR-amplified from the plasmid used in our previous study with the same primers (Appendix A) and same restriction enzyme cloning strategy [21]. Following PCR amplification and restriction digestion, wild-type Ku70 was cloned into the BioID2 plasmid, MCS-BioID2-HA (Addgene).

The Ku70 NLS (amino acids 539–556) was cloned into the empty BioID2 plasmid. A double-stranded oligonucleotide, obtained by annealing two complementary single-stranded primers (Integrated DNA Technologies) containing the Ku70 NLS sequence (Appendix A) with BamHI overhangs on both ends, was cloned into the BamHI site of the empty BioID2 plasmid.

Following PCR amplification (using forward and reverse primers, Appendix A), the Ku70 vWA domain (amino acids 1–250) was sub-cloned to the N-terminal of the NLS-BioID2-HA fusion protein.

Finally, the Ku70 ∆vWA was cloned into the empty BioID2 plasmid by PCR amplifying the Ku70 ∆vWA region (amino acids 251–609) using forward and reverse primers (Appendix A). All clonings were verified by sequencing analysis at the DNA Sequencing Facility at Robarts Research Institute, London, Canada.

### 2.2. Cell Culturing, Transfections, Stable Cell Line Generation

HeLa and HEK293 cell lines (both purchased from ATCC, Manassas, VA, USA) were used in this study. Cells were cultured in high-glucose Dulbecco’s modified Eagle’s medium (DMEM) (Wisent Bioproducts, St. Bruno, QC, Canada) supplemented with 10% fetal bovine serum (FBS) (Wisent Bioproducts) at 37 °C in 5% CO_2_. HeLa cells were initially transiently transfected with the BioID2 plasmid constructs using jetPRIME transfection reagent (Polyplus-transfection, Illkirch-Graffenstaden, France), following the manufacturer’s instructions, and either harvested after 24 h for immunoblotting or fixed and permeabilized for immunofluorescence.

For stable cell line creation, 48 h after transient transfection using jetPRIME, HEK293 cells were selected for approximately 8 days in 450 μg/mL geneticin (G418) until “islands” of resistant, monoclonal cells were observed. Upon colony formation, monoclonal cells were isolated, grown, and screened for stable expression by Western blotting. Monoclonal cells expressing the fusion proteins were pooled together to create a heterogeneous, polyclonal stable cell line in which the majority of cells expressed the fusion proteins. The stable cell lines were henceforth constantly maintained in 450 μg/mL G418.

### 2.3. Indirect Immunofluorescence

Cells were seeded onto coverslips, fixed using 4% paraformaldehyde (PFA), then permeabilized with 0.5% Triton X-100 for 15 min at room temperature (RT). Cells were incubated overnight at 4 °C with primary mouse antibody against HA (H9658, Sigma-Aldrich, Oakville, ON, Canada, 1:1000) and then with secondary Alexa 488 antibody against mouse (Invitrogen, Life Technologies, Burlington, ON, Canada, 1:1000). The coverslips were mounted using ProLong Gold containing 4′,6′-diamidino-2-phenylindole (DAPI) (Invitrogen). Cell images were taken with an Olympus BX51 microscope at 40× magnification and Image-Pro Plus (v5.0) software (Media Cybernetics, Inc., Bethesda, MD, USA).

Mean nuclear intensities for immunofluorescence images of each BioID2 construct were determined using ImageJ (v2.1.0). For Regions of Interest (ROI), area and mean pixel intensities (scored between 0 and 255) were acquired by overlaying the binary DAPI image outline mask onto the HA fluorescence image and then averaging the mean ROI intensities and subtracting the background signal acquired from the HEK293 negative control cell line.

### 2.4. Preparation of Extracts and Immunoblotting

Whole cell extracts (WCE) were prepared either using WCE buffer or RIPA lysis buffer, as previously described [21]. Nuclear extracts were prepared as previously described [89]. Extracts were resolved by SDS-PAGE (10%) before transfer onto a polyvinylidene difluoride (PVDF) membrane and blocking in 5% skim milk in TBST solution. Membranes were hybridized overnight with the following antibodies: mouse anti-HA (H3663, Sigma-Aldrich, 1:1000), goat anti-Ku80 (M-20; Santa Cruz, Dallas, TX, USA, 1:250), rabbit anti-RNF113A (HPA000160, Sigma-Aldrich, 1:250), rabbit anti-Spindly (24689-1-AP, Proteintech, Rosemont, IL, USA, 1:2000), and mouse anti-α-tubulin (T5168, Sigma-Aldrich, 1:1000). Biotinylated proteins were detected similarly: following transfer, PVDF membranes were blocked overnight in 5% bovine serum albumin in TBST solution and incubated for 1 h with horse radish peroxidase (HRP)-conjugated streptavidin (Pierce^TM^ High Sensitivity Streptavidin-HRP, ThermoFisher Scientific, Waltham, MA, USA, 1:20,000). Blots were imaged using the Clarity Western ECL substrate (Bio-Rad, Hercules, CA, USA) and the Molecular Imager^®^ ChemiDoc^TM^ XRS system (Bio-Rad) with Image Lab (v6.0.1.).

### 2.5. Co-Immunoprecipitation of Ku80, RNF113A, and Spindly

For the Ku80 co-immunoprecipitation, the established stable cell lines were used to produce 1 mg WCE that was diluted to 0.25% NP-40 and pre-cleared for 30 min before incubation at 4 °C overnight with the primary anti-HA antibody (H3663, Sigma-Aldrich). Immunoprecipitated proteins were next isolated using Pierce^TM^ Protein G magnetic beads (ThermoFisher Scientific, Rockford, IL, USA) and washed thrice (Wash buffer composition: 60 mM KCl, 25 mM HEPES pH 7.9, 0.5 mM ETDA pH 8, 0.5% NP-40, 12% glycerol) before being analyzed by Western blotting.

For the RNF113A and Spindly co-immunoprecipitation, 500 μg of nuclear extracts were diluted to 0.1% NP-40 and 100 mM KCl, pre-cleared, then incubated with primary anti-HA antibody (H3663, Sigma-Aldrich) overnight. Immunoprecipitated proteins were isolated and washed the same way as for the Ku80 co-immunoprecipitation before analysis by Western blotting.

### 2.6. Double-Stranded DNA (dsDNA) Pull-Down Assay

This assay was done exactly as previously described [21].

### 2.7. Biotinylation and Streptavidin Pull-Down of Biotinylated Proteins

For biotinylation, five 15 cm plates of each stable HEK293 cell line were grown to 85–95% confluency and then supplemented with 50 μM filter-sterilized biotin and incubated at 37 °C in 5% CO_2_ for 24 h to allow for sufficient biotinylation.

Cells were harvested then lysed for WCE at RT using RIPA lysis buffer (0.1% SDS, 0.5% sodium deoxycholate, 1% NP-40, 50 mM Tris-HCL, 150 mM NaCl, supplemented with protease inhibitors 0.2 mM phenylmethane sulfonyl fluoride (PMSF), 1 mM dithiothreitol (DTT), 1 g/mL leupeptin, 10 g/mL aprotinin, 1 g/mL pepstatin (inhibitors obtained from BioShop, Burlington, ON, Canada). Cell extracts were then sonicated (3 short bursts for 5 s at 30% vibration amplitude with 10 s intervals for cooling) and treated with Benzonase^®^ Nuclease (Sigma-Aldrich).

Samples were pelleted (4 °C at 17,968× *g* for 30 min) and the supernatant was collected and incubated overnight with magnetic Dynabeads (MyOne Streptavidin C1; Invitrogen) to bind biotinylated proteins. Protein-bound beads were collected and washed once with Strep-biotin wash buffer (50 mM Tris-HCL pH 8, 1% SDS (*w*/*v*), 150 mM NaCl) at RT, rotating for 5 min. Beads were then washed twice with RIPA lysis buffer, followed by three washes in TAP lysis buffer (10% glycerol, 0.1% NP-40, 2 mM EDTA pH 8, 50 mM HEPES pH 7.9, 100 mM KCl). Finally, beads were washed three times using 50 mM NH_4_HCO_3_ (ammonium bicarbonate, ABC) pH 8.0 solution before resuspension in ABC solution.

### 2.8. On-Bead Protein Digestion

Proteins bound to the magnetic beads were treated with 1 M iodoacetamide (IAA) at RT. After, beads were held in place using a magnetic rack and the supernatant containing the IAA was removed. Beads were re-suspended in 50 mM ABC. Next, protein-bound beads were digested with mass spectrometry-grade Lysyl Endopeptidase^®^ (Lys-C) (125-05061, Wako Pure Chemical Ind., Osaka, Japan) at 37 °C for 4 h before digestion overnight (approximately 10–12 h) with Trypsin/Lys-C mix (V5071, Promega, Madison, WI, USA) at 37 °C, followed by a final 4 h digest using mass spectrometry-grade trypsin (V5111, Promega) at 37 °C. For all samples, the missed cleavage rate was, on average, 5.53% with the highest missed cleavage rate being 8.19% while the lowest was 3.95%.

Magnetic beads were held in place using a magnetic rack and the supernatant, containing the released peptides, was transferred to a fresh, low-retention 1.5 mL microcentrifuge tube. The beads were rinsed once with 100 μL of H_2_O and the 100 μL wash was also added to the fresh, low-retention 1.5 mL microcentrifuge tube while the beads were discarded. Samples were dried using an Eppendorf™ Vacufuge™ concentrator spinning at 60 °C for about 1 h before re-suspension in 20 μL 0.1% trifluoroacetic acid. Samples were desalted and concentrated using C18 ZipTips (Merck Millipore Ltd., Cork, Ireland) before drying samples using the Eppendorf™ concentrator again spinning at 60 °C for about 15 min and re-suspending in 11 μL 0.1% formic acid.

### 2.9. Liquid Chromatography Electrospray Ionizing Tandem Mass Spectrometry (LC-ESI-MS/MS) and Data Analysis

Following the on-bead digestion, re-suspended samples were quantified using a NanoDrop^TM^ 2000/2000c (ThermoFisher Scientific) and submitted to the UWO Biological Mass Spectrometry Laboratory/Dr. Don Rix Protein Identification Facility (London, CA) for analysis of peptides by high-resolution LC-ESI-MS/MS using an ACQUITY M-Class UHPLC system (Waters) connected to an Orbitrap Elite mass spectrometer (ThermoFisher Scientific). Solution A consisted of water/0.1% formic acid (FA). Solution B was acetonitrile (ACN)/0.1% FA. Peptides (~1 μg) were injected onto an ACQUITY UPLC M-Class Symmetry C18 Trap Column, 5 μm, 180 μm × 20 mm, and trapped for 6 min at a flow rate of 4 μL/min at 99% Solution A/1% Solution B. Peptides were separated on an ACQUITY UPLC M-class Peptide BEH C18 Column, 130 Å, 1.7 μm, 75 μm × 250 mm, operating at a flow rate of 300 nL/min at 35 °C using a non-linear gradient consisting of 1–7% Solution B over 1 min, 7–23% Solution B over 179 min and 23–35% Solution B over 60 min before increasing to 95% Solution B and washing. Samples were run in positive ion mode.

Data was acquired using an FT/IT/CID Top 20 scheme with lock mass. Data was processed using PEAKS Studio version 8.5 (Bioinformatics Solutions Inc., Waterloo, ON, Canada), using 3 missed cleavages and semi-specific cleavage. Parent mass error tolerance was set to 20 ppm, fragment mass error tolerance was set to 0.8 Da. Protein and peptide False Discovery Rate (FDR) was set to 1%. Cysteine carbamidomethylation was set as a fixed modification while oxidation (M) and N-terminal deamidation (NQ) were set as variable modifications (maximum number of modifications per peptide = 5). All raw MS files were searched in PEAKS Studio version 8.5 using the Human Uniprot database (reviewed only; updated November 2019). In terms of result filtration parameters, proteins were identified using a minimum of ≥1 unique peptide(s) with the FDR set to 1%. For simplicity, the majority of proteins are referenced by their gene name. The mass spectrometry proteomics data have been deposited to the ProteomeXchange Consortium [90] via the PRIDE [91] partner repository with the dataset identifier PXD020165 and 10.6019/PXD020165.

### 2.10. SAINTexpress Analysis

Scaffold v4.8.7 (Proteome Software Inc., Portland, OR, USA) was used to validate the MS/MS-based peptide and protein identifications, based on the Peptide Prophet algorithm [92] with Scaffold delta-mass correction and the Protein Prophet algorithm [93], respectively. Peptides had to be identified with at least 95% probability, and only proteins identified with a minimum 95% probability (resulting in a protein FDR < 1%), using at least two unique peptides, were analyzed further. Spectral counts were exported from Scaffold for all four stable HEK293 cell lines and the HEK293 control and formatted for SAINTexpress analysis [15], a computational algorithm integrated into the REPRINT website (available at https://reprint-apms.org (accessed on 14 March 2021)). Proteins with a SAINTexpress score ≥0.6 were incorporated into a network figure, highlighting the high confidence interactors scoring ≥0.9 and ≥0.99. Network mapping information was exported from REPRINT and visualized with Cytoscape (v3.6.1).

### 2.11. Duolink^®^ In situ Proximity Ligation Assay (PLA)

Cells were seeded, fixed, permeabilized, and blocked as described for immunofluorescence. Following blocking, cells were incubated overnight at 4 °C with both mouse and rabbit antibodies. The primary mouse antibodies used were against either HA (H9658, Sigma-Aldrich, 1:1000) or Ku70 (N3H10, Santa Cruz, 1:1000). Rabbit primary antibodies used were for Ku80 (H-300, Santa Cruz, 1:1000), RNF113A (HPA000160, Sigma-Aldrich, 1:100), or Spindly (24689-1-AP, Proteintech, 1:1000). Following primary incubation, PLA was conducted by following the manufacturer’s instructions (Sigma-Aldrich). Probes used were anti-Mouse MINUS and anti-Rabbit PLUS.

Cells were mounted as described for immunofluorescence and images were taken with an Olympus BX51 microscope at 40× magnification (using the CY3 channel) and Image-Pro Plus (v5.0) software (Media Cybernetics, Inc.).

Quantitative analysis determining the number of PLA foci per cell was determined using ImageJ (v2.1.0). Binary DAPI images were used to highlight each nucleus (size set to 0.003-infinity) as a ROI. Similarly, binary images outlining the PLA foci (size set to 0.0003-infinity) were also acquired and merged with the DAPI outline. The number of PLA foci were plotted as bars indicating mean number of nuclear PLA foci and with dots indicating the exact number of nuclear foci per cell (n = 65 cells, from at least 2 separate PLA experiments were counted for each PLA condition) with 95% confidence intervals indicated. The negative control (Ku70 only) mean was compared to Ku70 with Ku80, RNF113A, or Spindly using unpaired, two-tailed t-tests assuming unequal variance.

## 3. Results and Discussion

### 3.1. Designing the Ku70-BioID2 Constructs

Four Ku70 fusion constructs were designed, with each being fused to the mutant biotin ligase (denoted BioID2, representing the specific biotin ligase used to conduct BioID2). Our design process was guided by our prior Ku70-BioID study [21], thus the BioID2 enzyme was added to the C-terminus of the constructs in this study. The biotin ligase contains a C-terminal HA tag that was used in subsequent experiments.

Each construct contained different segments of Ku70, all specifically centered on identifying vWA domain-specific interactors (Table 1). As Ku is established as a nuclear protein with each subunit having its own nuclear localization signal (NLS) in the C-terminal region [94,95,96], we added the Ku70 NLS for constructs lacking a means to transport themselves into the nucleus. After completing the initial cloning experiments, the NLS and Ku70 vWA-NLS BioID2 constructs were transiently transfected into HeLa cells and imaged using indirect immunofluorescence to ensure that efficient nuclear localization with the Ku70 NLS was observed (Appendix A).

The NLS construct represents a negative control in which a nuclear-targeted BioID2 is used to identify and subtract non-specific protein interactors. Meanwhile, full-length Ku70 represents a positive control that should theoretically identify all Ku70 interactors. Ku70 vWA is an experimental construct that only contains the Ku70 vWA domain and is also targeted specifically to the nucleus by the intrinsic Ku70 NLS. Finally, the Ku70 ∆vWA does not contain the vWA domain and represents an experimental construct that should be mutually exclusive to the Ku70 vWA construct. The composition of each construct is depicted as a schematic (Figure 2a).

### 3.2. Creating Ku70-BioID2 Polyclonal Stable Cell Lines in HEK293 Cells

In order to identify Ku70 vWA-specific interactors, we created four stable cell lines that constitutively expressed our BioID2 constructs in HEK293 cells. We used the HEK293 cell line specifically because we had already conducted BioID previously in the same cell line [21]. Following transfection, G418-resistant monoclonal colonies were isolated and screened for expression by Western blotting. Clones demonstrating stable expression were pooled for each of the constructs and tested for expression again by Western blotting (Figure 2b).

To check that the majority of cells were expressing the Ku70-BioID2 fusion proteins and to confirm the nuclear localization of each construct, we used indirect immunofluorescence microscopy to detect the HA tag at the C-terminus of each fusion protein. From the immunofluorescence, we were able to confirm that the majority of cells in our polyclonal stable cell lines were expressing the fusion proteins and we observed efficient nuclear localization, suggesting the Ku70 NLS was effective at nuclear targeting (Figure 2c). Curiously, unlike the NLS and full-length Ku70 stable cell lines, some weak cytoplasmic staining was observed in the Ku70 vWA and Ku70 ∆vWA stable cell lines. These differences could be due to the conformation of some fusion proteins partially obstructing the NLS from being recognized by importins, which allow proteins passage into the nucleus [97]. However, the mean nuclear fluorescence intensities for each construct were determined using ImageJ and were shown to be highly similar between constructs (Appendix A), indicating no significant differences in nuclear expression between fusion proteins.

### 3.3. Biotinylation and Expression of Stable Cell Lines

To ensure that the biotin ligase in each of the fusion proteins was capable of biotinylating proteins, the stable cell lines were incubated with or without 50 μM supplemental biotin for a 24-h incubation period to acquire sufficient protein biotinylation. Cells were harvested and lysed for whole cell protein extracts, and extracts were run via SDS-PAGE and visualized by Western blotting (Figure 3a). Biotinylation was evident in all lanes except in the negative control HEK293 cell line. As expected, a greater extent and intensity of biotinylation was observed in lanes with supplemental biotin. More biotinylated proteins were detected with the NLS-BioID2 than with the other constructs, as evidenced by the greater number of bands (Figure 3a, lane 3). This greater level of biotinylation could be attributed to protein expression differences between the fusion proteins (Figure 3a), the different construct-specific contexts where interactions may be occurring, or due to differences in the tagging flexibility of the biotin ligase enzyme attached to each different-sized bait. Importantly, the Ku70 vWA-BioID2 and Ku70 ∆vWA-BioID2 appeared to be expressed at similar levels and have a similar biotinylation efficiency, which means that biotinylation by these two fusion constructs can be directly compared.

### 3.4. Co-Immunoprecipitation with Ku80 and Association with Broken Double-Stranded DNA

Aside from testing localization and biotinylation, we also wanted to ensure that the biotin ligase addition did not prevent the heterodimerization of Ku70 and Ku80, nor Ku’s association with double-stranded DNA (dsDNA) ends. Therefore, we conducted Ku80 co-immunoprecipitation and dsDNA pull-down experiments with our HEK293 stable cell lines. As these experiments had been conducted successfully during our earlier Ku70-BioID study, we expected that the switch to BioID2 would not alter the results. We assessed all the different Ku70 segment constructs for association with Ku80 and dsDNA.

Both full-length Ku70 and the Ku70 ∆vWA fusion proteins co-immunoprecipitated successfully with Ku80 (Figure 3b), suggesting that both of these fusion proteins are capable of forming a heterodimer. This was within our expectations as the Ku70 region required for Ku80 heterodimerization is located within the DNA-binding core domain [98], thus the vWA domain is not necessary for Ku80 association. Correspondingly, the NLS and Ku70 vWA fusion proteins alone were incapable of Ku80 heterodimerization. While this was expected, it raises an important issue. As Ku is believed to be an obligate heterodimer, using Ku70 domains incapable of heterodimerization may either produce false positive protein interactions and/or miss key interactions that only occur with the full heterodimer. At the same time however, issues associated with studying isolated protein domains have been noted previously [99,100,101] and should not prevent the exploration of the BioID2 technique in this manner.

Using a DNA pull-down with a short dsDNA probe, we also tested whether the fusion proteins were capable of associating with broken dsDNA. Theoretically, if the fusion proteins formed a heterodimer with Ku80, they should associate with dsDNA ends in a sequence-independent manner [102]. Both full-length Ku70 and the Ku70 ∆vWA fusion proteins were pulled down with the dsDNA probe (Figure 3c), implying that these fusion proteins formed a heterodimer with Ku80 and could consequently associate with dsDNA ends.

### 3.5. Conducting BioID2 and Preliminary Analysis

Having established proper expression, localization, and activity for all constructs, we proceeded with the BioID2 experiment. Following the incubation of the four cell lines with biotin for 24 h, extracts were prepared, and biotinylated proteins were isolated using streptavidin beads. Three biological replicates were conducted for each of the established cell lines, and isolated biotinylated proteins were analyzed by liquid chromatography-electrospray ionization-tandem mass spectrometry (LC-ESI-MS/MS). The results for all three MS replicates were pooled for each of the cell lines. In addition to the four cell lines, the HEK293 cell line (without BioID2) was also included in the study as an additional negative control, identifying proteins that interacted non-specifically with the streptavidin beads. On average, a total of 756 proteins were identified for the HEK293 cell line, 1035 for NLS-BioID2, 814 for Ku70-BioID2, 1053 for Ku70 vWA-BioID2, and 877 for Ku70 ∆vWA-BioID2 (Figure 4a).

Comparing the total proteins identified in the two negative control cell lines of HEK293 and NLS-BioID2, we observed some overlap in the proteins identified (769 proteins, or 34.5%), but almost 50% of the proteins identified were exclusive to NLS-BioID2 while about 20% were exclusive to HEK293 (Figure 4b). This comparison was valuable because it demonstrates that the NLS-BioID2 alone does not fully encompass the proteins that may interact non-specifically with the streptavidin beads. In other words, having multiple negative controls is particularly useful for filtering out non-specific protein interactors, frequent MS background or contaminating proteins, and potential false positives.

The total unfiltered unique proteins identified in any of the three biological MS replicates for the three Ku70-BioID2 cell lines (Ku70, Ku70 vWA, and Ku70 ∆vWA) were compared and 804 proteins or roughly 36% of the total identified proteins were shared between the three cell lines (Figure 4c). At this stage, since the negative control proteins have yet to be subtracted from the candidate interactors, this data represents the unfiltered results.

In almost every sample, all five carboxylase enzymes (pyruvate carboxylase/PC, acetyl-CoA carboxylase 1/ACACA and 2/ACACB, propionyl-CoA carboxylase α chain/PCCA, and methylcrotonoyl-CoA carboxylase subunit α/MCCCA) that are naturally biotinylated in human cells were identified. The presence of the carboxylases in the MS results acts as an innate positive control for biotinylation and confirms that biotinylated proteins were successfully captured [1].

### 3.6. Filtering, Comparing, and Processing Ku70 vWA- and ∆vWA-Specific Protein Interactors

To filter out false positive proteins, non-specific protein interactors that appeared in at least two biological replicates of either the HEK293 or NLS-BioID2 cell lines were subtracted from the full-length Ku70-, Ku70 vWA-, and Ku70 ∆vWA-BioID2 results. Since numerous proteins were identified in the negative controls, some of which included known Ku interactors, we wanted to avoid over-penalizing true interactors [103]. Additionally, we also wanted to see some consistency in the appearance of the non-specific proteins. For these reasons, we required that a candidate must appear in at least two replicates of the negative controls in order to be subtracted (Figure 5a). In addition to the negative control proteins, any remaining keratin proteins were also removed, as keratin was likely a contaminant introduced during sample preparation.

After filtering the candidates to remove the negative control proteins, the filtered candidates for the different Ku70-BioID2 constructs were compared again (Figure 5b). The specific proteins identified (appearing in one or more biological replicate) in each segment of the Venn diagram were noted (Appendix A). Though we also explored alternative ways to compare the candidates for the different Ku70 constructs by comparing proteins that appeared in two or more replicates (*n* ≥ 2, Appendix A) or three replicates (*n* = 3, Appendix A, Appendix A), we ultimately deemed the following three criteria as essential to identifying Ku70 vWA- or Ku70 ∆vWA-specific candidates:Since Ku70 vWA and ∆vWA are mutually exclusive, no true BioID2 candidates would be shared, irrespective of the number of biological replicates in which the candidate appears;Both Ku70 vWA and ∆vWA candidates should be shared with full-length Ku70, meaning the candidate must be present in at least one biological replicate of Ku70-BioID2, which represents a positive control;All final candidates must appear in all three biological replicates of either Ku70 vWA- or Ku70 ∆vWA-BioID2 results.

Comparing the total, negative-control filtered proteins between the three experimental cell lines (Ku70-, Ku70 vWA-, and Ku70 ∆vWA-BioID2), 8.2% of the total identified proteins were shared between all three cell lines (Figure 5b). This 8.2% represents 97 proteins, which is a substantial reduction from the unfiltered value of 804 proteins (Figure 4c), implying the filtering process was generally effective at removing most background MS proteins present in all three cell lines. However, as the Ku70 vWA- and Ku70 ∆vWA-BioID2 results should be mutually exclusive, the 97 proteins shared between all 3 constructs can be treated as likely false positives (Figure 5b).

Similarly, exactly 114 proteins (9.6%) were found to be common between Ku70 vWA- and Ku70 ∆vWA-BioID2 (Figure 5b). Again, as these baits should be mutually exclusive, these remaining proteins also most likely represent false positives rather than true interactors. Indeed, many of these candidates are highly expressed proteins involved in cell structure, mitochondrial function, or RNA splicing that likely represent frequent MS contaminants rather than true interactors (Appendix A). For example, one of the candidates, PALLD, encodes a cytoskeletal protein needed for the organization of the actin cytoskeleton and is found abundantly throughout the cell [104]. The number of mutually exclusive false positive candidates between Ku70 vWA and ∆vWA could be further reduced by changing the comparison criteria (i.e., only comparing candidates that appeared in two or more or three replicates) (Appendix A), though this could not be reduced to zero.

Interestingly, 63 proteins (5.3%) were shared between full-length Ku70 and the Ku70 vWA domain, while 102 proteins (8.6%) were common to full-length Ku70 and Ku70 ∆vWA (Figure 5b). Overall, close to 50% more proteins were identified in the Ku70 vWA- and Ku70 ∆vWA-BioID2 cell lines than full-length Ku70.

One possible explanation for why Ku70 vWA and ∆vWA appear to have more candidate proteins identified than full-length Ku70 could be that the biotin ligase is less hindered in the first two cell lines and may be more sterically free when attached to smaller bait proteins. For example, because the Ku70 vWA represents only a small protein domain, it may be less restrained in terms of biotinylating its targets compared to the full-length protein. This could suggest that many of the candidate interactors are artificial in the sense that Ku70 may not normally interact with such proteins. Another explanation is that these two Ku70 domain constructs are smaller and expressed at higher levels, which could be contributing to the increased number of interactors identified (see Figure 3a).

### 3.7. Identifying and Analyzing Ku70 ∆vWA-Specific BioID2 Candidate Proteins

To identify candidate proteins that interacted with regions other than the Ku70 vWA domain, factors from the Ku70 ∆vWA-BioID2 list that were shared with full-length Ku70 and absent from those identified with the Ku70 vWA construct were selected (Appendix A). From this list, 7 candidate proteins appeared in three biological replicates (Table 2).

Since the Ku70 ∆vWA is still capable of Ku heterodimerization, which requires the Ku70 C-terminus [98], we expected to see Ku80-specific interactors using this construct with BioID2. Accordingly, we identified APLF as a Ku70 ∆vWA-specific candidate protein that appeared in all three biological replicates (Table 2). APLF contains a KBM and is known to be a direct Ku80 interactor [58]. Aside from APLF, we also observed WRN and NKRF, both previously identified as being Ku interactors [28,57]. While the exact interaction of NKRF with Ku has not been mapped to either subunit, WRN has been proposed to bind both Ku subunits, and its region of interaction within Ku70 has been mapped to the C-terminus, from residues 542 to 609 [57]. The exclusive identification of APLF and WRN in the Ku70 ∆vWA-BioID2 MS results, suggests that the region-mapping approach was successful.

Other known Ku interactors were not identified definitively using this segmented Ku70 approach. This could be due to overly stringent filtering criteria or that the negative controls (in particular, the NLS-BioID2) were too promiscuous in tagging. For example, although MRE11A is well-established as a direct interactor of the Ku70 C-terminal region [62], it appeared in one replicate of the HEK293 negative control cell line and in all three replicates of NLS-BioID2 and was thus subtracted. This indicates that even proteins that are true interactors can be dismissed as non-specific interactors. Notably, MRE11A was observed in every replicate of the BioID2 results for all of the Ku70 constructs, including Ku70 vWA with which it was not expected to interact. The appearance of MRE11A in both Ku70 vWA and Ku70 ∆vWA BioID2 datasets implies that some of the identified proteins, including known interactors, can confound domain-mapping attempts. In the future, deciphering the true interactors from the numerous false positives will need to be addressed as this appears to be one of the greatest drawbacks for the BioID technique and its newer variations [105]. The issue of false positives was made particularly evident here by using different segments of the same bait protein to map protein interactions.

TERF2IP (also known as Rap1), a telomere-specific protein, is another example of a known Ku interactor, though this interaction has yet to be mapped to a specific Ku subunit or domain [24]. TERF2IP was not observed in our previous Ku70-BioID study [21], yet it was observed in our current study in one biological replicate of Ku70 vWA-BioID2 but was absent from the full-length Ku70-BioID2 results. Due to its absence from the full-length Ku70 results and because it did not appear consistently in 3 replicates of Ku70 vWA-BioID2, it did not reach the final candidate list based on our stipulated criteria.

Certain protein interactions may be infrequent and therefore more difficult to capture; this could offer an explanation as to why TERF2IP only appeared in one MS replicate of the Ku70 vWA-BioID2 screen and was absent from the full-length Ku70-BioID2 results. The overall lower expression of the full-length Ku70-BioID2 compared to Ku70 vWA-BioID2 could provide another possible explanation for why candidates are not observed in the full-length Ku70 results. It is also possible that there may be stronger steric hindrance of the biotin ligase attached to full-length Ku70 compared to the shorter constructs, which may account for differences in the protein candidates identified. Notably, proteins must also have accessible lysine residues in order to be successfully tagged by the biotin ligase [1,2], thus candidates with fewer accessible lysine residues may not be as readily identified.

Known Ku70 interactors may have been missed in our BioID2 results due to the inherent nature of their interaction with Ku70. Some protein interactions have been shown to be reliant on the presence of DNA, such as XLF [26] and REF1 [27]. Other interactions are cell-type specific, including DNTT/TdT [25] and VAV1/p95^vav^ [63]. In particular, the addition of the biotin ligase to the C-terminus of Ku70 could be detrimental to the identification of some Ku70 SAP (SAF-A/B, Acinus, and PIAS) domain interactors. The biotin ligase may have masked the site of interaction, preventing some proteins from interacting or potentially excluding them from biotin-tagging. Less abundant interactions produce fewer biotin-tagging events, making it more challenging to detect these tagged proteins. While the abundance and frequency of protein interactions are useful indicators for identifying putative protein interactors, not every tagged protein ends up being detected or identified. One potential way to overcome this would be to attempt BioID2 with fewer proteins. For example, using sub-cellular fractionation to separate cells into multiple compartments (i.e., mitochondria, nucleoplasm, chromatin-associated, cytoplasm, etc.) might allow for the detection of less abundant interactors.

### 3.8. Identifying and Analyzing Ku70 vWA-Specific BioID2 Candidate Proteins

Comparing the filtered candidates as described above, we were able to identify Ku70 vWA- and Ku70 ∆vWA-specific proteins (Appendix A). Of these candidates, eleven proteins were found to appear in three biological replicates of Ku70 vWA-BioID2 (Table 3). These candidates were shared with full-length Ku70-BioID2 while being absent from the Ku70 ∆vWA-BioID2 results, as stipulated by the three criteria described earlier.

Given that only a few proteins have been reported to actually interact directly with the Ku70 vWA domain, it was unsurprising that none of the identified candidate proteins were known Ku70 vWA interactors (Figure 1). Many of the candidates identified were predominately nuclear proteins (Table 3). One of the candidates, YLPM1, is involved in regulating levels of TERT, the catalytic subunit of telomerase [106], and appeared in our earlier BioID study [21]. Another candidate, ZNF830 (also referred to as OMCG1), is an essential gene that acts as a cell cycle regulator by maintaining genome integrity through various means [107,108]. For example, in mouse embryonic fibroblast cells, ZNF830 deficiency was shown to impair cell survival due to DNA replication fork collapse, dsDNA break formation, and cell cycle checkpoint activation [107]. Most recently, another group showed that ZNF830 binds to dsDNA breaks, directly interacts with CtIP, and regulates CtIP recruitment to DSBs [108].

A third Ku70 vWA interaction candidate, SPDL1 or Spindly, has been shown to localize to the kinetochore and mitotic spindle poles and has been implicated in recruiting motor proteins to the kinetochore [109]. A fourth candidate identified here includes RNF113A, an RNA-binding protein previously known to regulate splicing [110].

In our previous BioID study, which used Ku70 as the bait protein, we could not exclude Ku80 interactors from our results and could therefore only classify our candidates as Ku interactors, collectively [21]. One of the advantages of using the Ku70 vWA domain to study protein interactions is that the vWA domain alone cannot form a heterodimer with Ku80, thus excluding Ku80 interactors. None of the identified Ku70 vWA interactors that appeared in any of the replicates were known Ku80 interactors (Appendix A), suggesting the experimental design excluded these interactors.

In general, the Ku70 vWA-BioID2 candidates were challenging to interpret as protein interactions with the vWA domain may be conditional [111], dependent upon context, or reliant on specific post-translational modifications [89]. As of yet, it is still unclear why so few interactions have been mapped to the Ku70 vWA domain.

### 3.9. Validating RNF113A and Spindly as Ku70 and Ku70 vWA Proximal Interactors

In order to confirm interaction with the Ku70 vWA domain, co-immunoprecipitation experiments using the Ku70 vWA-BioID2 and Ku70 ∆vWA-BioID2 HEK293 stable cell lines were attempted with candidates Spindly and RNF133A. Ku80 was used as a control as it would be expected to interact with Ku70 ∆vWA-BioID2 and not Ku70 vWA-BioID2. Despite attempting multiple conditions for binding, none of the co-immunoprecipitation experiments demonstrated a Ku70 vWA-specific interaction (Figure 6a, and unpublished results). However, while Spindly did not appear to co-immunoprecipitate, RNF113A did seem to co-immunoprecipitate with Ku70, albeit with both the Ku70 vWA and Ku70 ∆vWA constructs, suggesting it could be a non-specific interaction under in vitro conditions (Figure 6a).

To confirm that the RNF113A was biotinylated only in the Ku70 vWA-BioID2 stable cell line, the whole cell protein extracts that were prepared for MS were used to conduct a small-scale (1 mg) streptavidin pull-down (same steps as the large-scale MS pull-down procedure) that was probed for RNF113A (Figure 6b). The streptavidin pull-down confirmed that RNF113A was only biotinylated in the Ku70 vWA-BioID2 cell line (Figure 6b). We also checked to see whether Spindly could be seen using the streptavidin pull-down, however this candidate was not detected in the pull-down (Figure 6b). We next sought to determine if the RNF113A and Spindly interactions with the Ku70 vWA domain could be transient or weak, rather than constitutive and strong. BioID is a powerful technique for visualizing the context and landscape of protein function because it allows the identification of weak and transient interactions, but these cannot be easily verified by more stringent techniques such as co-immunoprecipitation. Therefore, we used proximity ligation assays (PLA) [112], a technique other BioID-based studies have used as a validation follow-up to BioID screening [113,114] because it allows for the detection of proteins complexes often not detected by co-immunoprecipitation.

Using PLA, we were able to confirm the co-localization of RNF113A and Spindly with the constitutively expressed Ku70 vWA-BioID2 construct in HEK293 cells while little to no signal was observed with the Ku70 ∆vWA-BioID2 construct (Figure 6c). To confirm that these interactions occurred in the context of endogenous proteins, we also conducted PLA in HeLa cells using an antibody against endogenous Ku70. These analyses showed a robust, nuclear PLA signal with endogenous Ku80, our positive control, and with both RNF113A and Spindly, suggesting complex formation or close proximity of Ku and these two candidate interactors (Figure 6d). The PLA signal for Ku70 with Ku80, RNF113A, and Spindly in HeLa cells was quantified to determine the average number of nuclear PLA foci per cell using ImageJ (Appendix A). The number of foci were significantly greater in Ku70 with either Ku80 (*p* = 2.50 × 10^−21^), RNF113A (*p* = 1.53 × 10^−15^), or Spindly (*p* = 3.97 × 10^−18^), compared to the Ku70 only control (Appendix A).

Very recently, another study has shown that RNF113A is recruited to sites of DNA damage and found that RNF113A is important for regulating DNA repair factor recruitment to chromatin [115]. This work also showed that RNF113A co-localized with phosphorylated DNA-PK_cs_ in cisplatin-treated cells [115], however as our study did not look at interaction in the context of double-stranded breaks, it is presently unclear why this protein may be found in proximity to Ku70. We have previously found that Ku appears to be in close proximity to many DNA repair factors and sensors, even in the absence of DNA damage [21]. Similarly, at present the context for Ku70 interaction with Spindly, a mitosis-related protein, is also unclear, although one study did observe differential Ku localization throughout mitosis [116], indirectly implying that Ku may also play a role in mitosis.

### 3.10. Candidate Identification Using SAINTexpress Analysis

For comparison, in addition to the filtering approach employed above, a second approach for selecting candidates was utilized. SAINTexpress is a quantitative algorithm developed specifically to filter and select high confidence candidates using quantification (in this case, spectral count data) [15]. Spectral counts were acquired for all negative controls and for the Ku70 ∆vWA- and Ku70 vWA-BioID2 cell lines. Candidate proteins with a SAINTexpress score ≥0.6 (Appendix A) were included in the candidate networks depicting Ku70 ∆vWA- and Ku70 vWA-BioID2 interactors (Appendix A). The networks also specifically highlight the high confidence interactors with a SAINTexpress score of ≥0.9 and ≥0.99 and whether or not there is previous evidence of interaction on the BioGRID database.

From the SAINTexpress analysis, we were able to identify APLF (SAINTexpress score = 0.99) and WRN (SAINTexpress score = 0.64) as high confidence Ku70 ∆vWA-BioID2 interactors (Appendix A). As these are known Ku interactors also found using our filtering approach (Table 2), this overlap suggests both filtering approaches are consistent. However, curiously, none of the high confidence interactors identified by SAINTexpress for Ku70 vWA-BioID2 were shared with those identified by our filtering approach, including RNF113A and Spindly that we validated through PLA (Table 3, Figure 6). While the reason for this is unclear at present, it suggests that the filtering process may need to be fine-tuned. Aside from SAINT and SAINTexpress, other algorithms (MiST [13], MUSE [16]) are also available for selecting interactors. Appropriate candidate filtering and selection remains one of the more challenging aspects in the BioID pipeline and, at present, there are multiple options from which to choose.

## 4. Conclusions

BioID is a powerful technique and represents a significant milestone in our ability to study protein-protein interactions within the cell. New variations of the BioID technique are frequently published and expand upon the existing potential of this technique. Here, we used BioID2 to identify and map domain- and region-specific interactions to a bait protein of interest, Ku70.

In the current study, using specific Ku70 constructs, we conducted BioID2 in HEK293 cells to identify Ku70 vWA- and ∆vWA-specific interactors. This is one of the few attempts that has been made with either BioID or BioID2 to directly map domain-specific interactions. Using this approach, we were able to identify protein candidates that interact with the Ku70 vWA domain and validated RNF113A and Spindly as proximal Ku70 interactors. Additionally, using the Ku70 ∆vWA cell line, we were able to identify some known C-terminal Ku70 interactors, suggesting that the bait protein segmentation approach can work with BioID2. At the same time, we also highlight the issue of false positives, demonstrated by the identification of proteins which appear to interact with both the Ku70 vWA domain and Ku70 ∆vWA, which should be mutually exclusive.

In sum, the segmented approach to BioID2 taken here appears to be feasible in identifying protein interactors. Incorporating multiple controls and using mutually exclusive domains or segments of a bait protein is a viable method for filtering the candidate proteins, albeit the negative controls and filtering criteria should be carefully determined beforehand. BioID2 can be used successfully with protein domains, suggesting it is possible to map the identified protein interactions to specific regions of a bait protein, but there are limitations and caveats that must be taken into consideration when designing such experiments.

## Figures and Tables

**Figure 1 cells-10-00646-f001:**
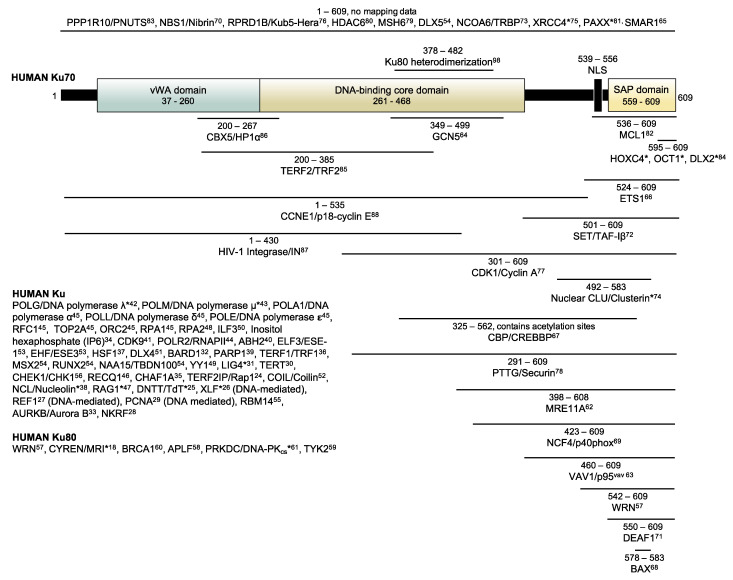
Schematic representation of Ku70 protein domains and mapped protein/molecule interactions. Interactions of the Ku80 subunit and Ku complex as a whole are listed. Note that interactions were determined using various cell lines and certain interactions may only occur under specific conditions. For example, TdT is only expressed in lymphocytes while VAV1/p95^vav^ is only expressed in hematopoietic cells. Proteins with an asterisk (*) only interact conditionally. Each cited interaction can be found in the References section through the indicated superscript number.

**Figure 2 cells-10-00646-f002:**
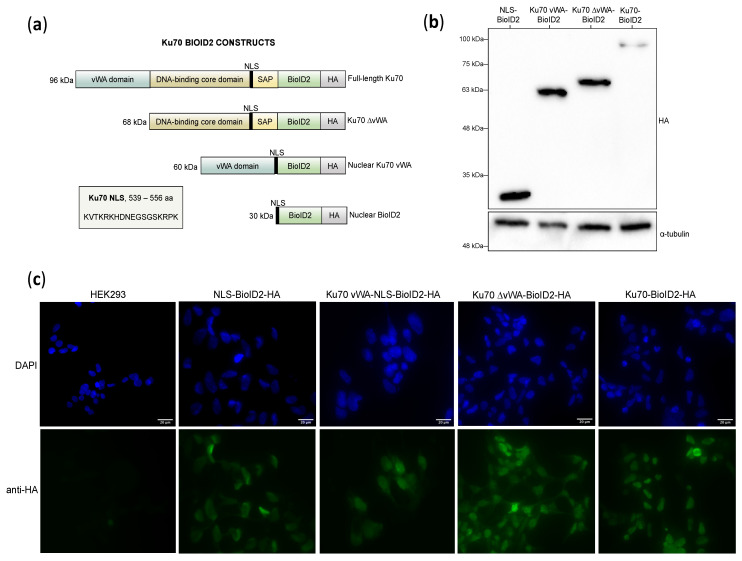
Testing Ku70-BioID2 constructs in HEK293 stable cell lines for expression and cellular localization. Note that all Ku70-BioID2 constructs are HA-tagged at their C-terminus: (**a**) Schematic diagram of Ku70-BioID2 constructs where NLS denotes nuclear localization signal and SAP denotes SAF-A/B, Acinus, and PIAS motifs domain; (**b**) Testing the expression of pooled, polyclonal HEK293 stable cell lines by Western blotting of 50 μg of whole cell extracts; (**c**) Indirect immunofluorescence microscopy of Ku70-BioID2 stable cell lines. Images taken at 40× magnification with white bar denoting 20 μm.

**Figure 3 cells-10-00646-f003:**
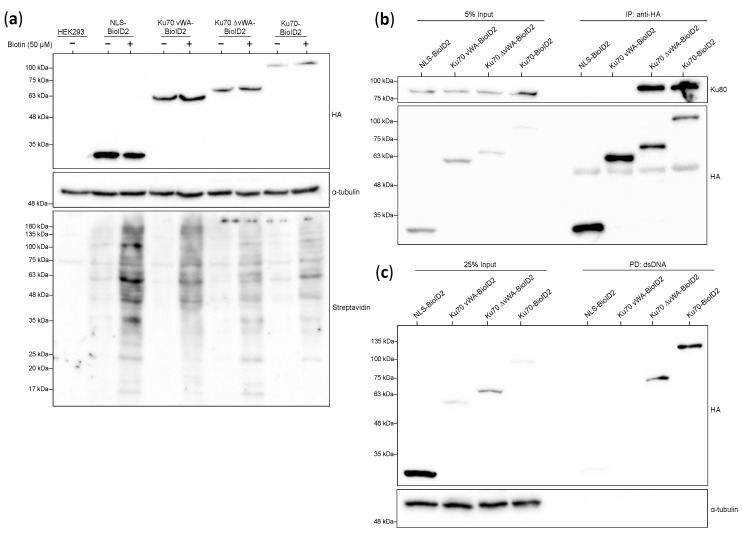
Preliminary experiments for Ku70-BioID2 constructs in HEK293 stable cell lines. Note that all constructs are HA-tagged at their C-terminus. Samples were visualized by Western blotting of 10% gels run by SDS-PAGE; (**a**) Testing biotinylation capability. Cell lines were incubated with or without 50 µM biotin for 24 h. Cells were lysed for whole cell protein extracts (WCE). Blots were probed with HRP-streptavidin; (**b**) Testing for Ku80 interaction. Co-immunoprecipitation conducted with 1 mg WCE; (**c**) Testing for DNA association. Double-stranded DNA pulldown conducted with 100 µg nuclear protein extracts.

**Figure 4 cells-10-00646-f004:**
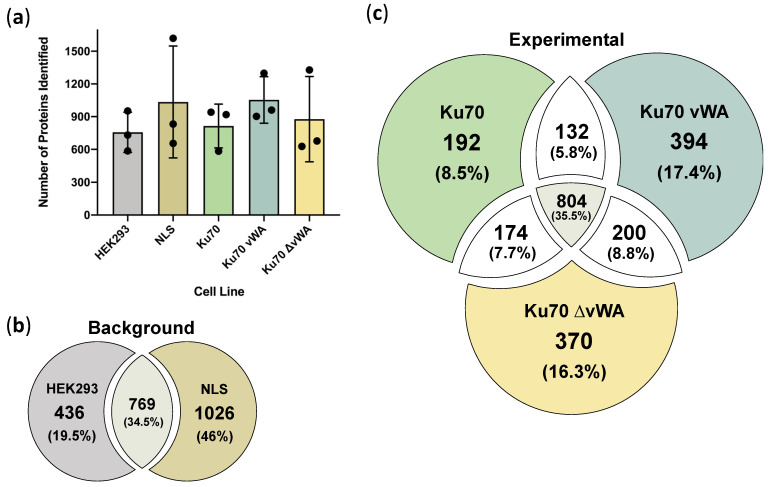
Comparing the total number of proteins identified by BioID2 using Liquid Chromatography-Electrospray Ionization-Tandem Mass Spectrometry (LC-ESI-MS/MS), where proteins were identified by having ≥1 unique peptide(s) and FDR was set to 1%; (**a**) Total number of proteins in three biological replicates of BioID2 in HEK293 stable cell lines: parental HEK293, NLS (Ku70)-BioID2, Ku70-BioID2, Ku70 vWA-NLS-BioID2, and Ku70 ∆vWA-BioID2. Each replicate is represented by a black dot; (**b**) Venn diagram of the total number of proteins for negative controls, HEK293 and NLS (Ku70)-BioID2. Any proteins that appeared at least once in the three biological replicates for each cell line are included; (**c**) Venn diagram of the total number of proteins identified for each experimental condition: full-length wild-type Ku70-BioID2, Ku70 vWA-NLS-BioID2, and Ku70 ∆vWA-BioID2. Any proteins that appeared at least once in the three biological replicates for each cell line are included. Subsequently, only candidate proteins that appeared in three biological replicates were selected for the final candidate lists.

**Figure 5 cells-10-00646-f005:**
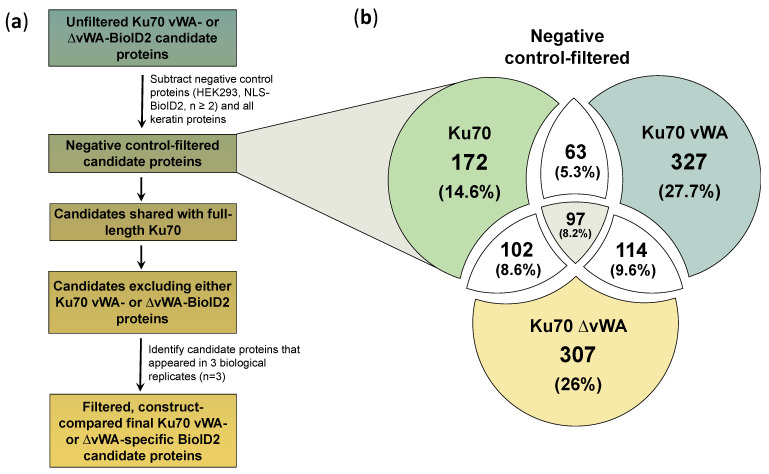
Filtering process and steps to compare filtered candidate BioID2 proteins; (**a**) Ku70 vWA-specific or Ku70 ∆vWA-specific candidates were first filtered by subtracting negative control proteins identified in two or more biological MS replicates (n ≥ 2) of either the HEK293 cell line or the NLS-BioID2 cell line. Additionally, all keratin proteins were removed. Next, all remaining candidates were compared to identify those that were shared with full-length Ku70 and missing from the mutually exclusive deletion construct (either Ku70 vWA or Ku70 ∆vWA). Finally, only candidates that appeared in all 3 biological replicates of Ku70 vWA or Ku70 ∆vWA were selected for further discussion and validation; (**b**) Venn diagram comparing the total number of negative-control and keratin filtered proteins for full-length wild-type Ku70-BioID2, Ku70 vWA-NLS-BioID2, and Ku70 ∆vWA-BioID2.

**Figure 6 cells-10-00646-f006:**
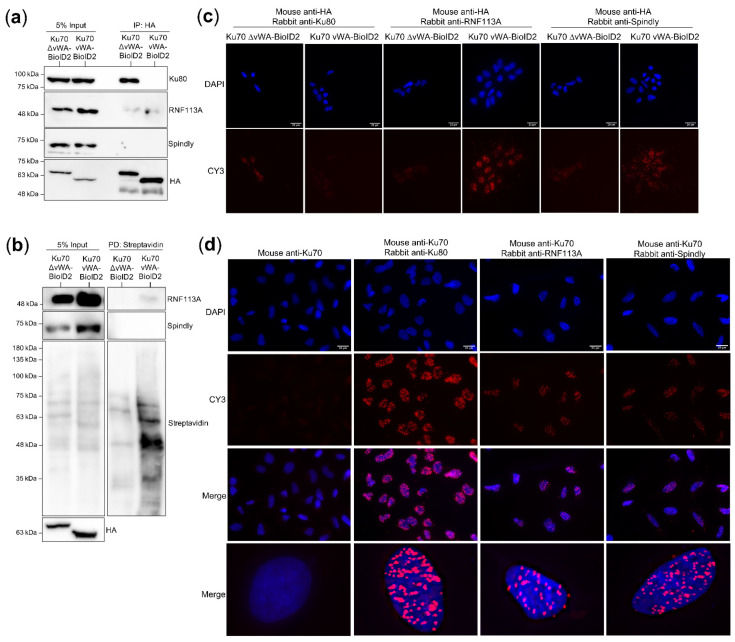
Validation testing of RNF113A and Spindly as Ku70 vWA-specific interactors; (**a**) Ku70 vWA-specific interactor nuclear protein extract (500 μg) co-immunoprecipitation (*n* = 3); (**b**) Small-scale streptavidin pull-down (1 mg) of biotinylated HEK293 whole cell extracts (*n* = 2, technical replicates); (**c**) PLA of Ku70 vWA- and Ku70 vWA-BioID2-HA stable HEK293 cell lines using mouse anti-HA and rabbit Ku80, RNF113A, or Spindly (*n* = 3); (**d**) PLA of HeLa cells detecting endogenous proteins using mouse anti-Ku70 and rabbit Ku80, RNF113A, or Spindly (*n* = 3). Images taken at 40× magnification with white bar denoting 20 μm. Bottom panels are enlarged images of nuclei selected from the panels above. Quantification analysis for the average number of nuclear PLA foci can be found in Appendix A.

**Table 1 cells-10-00646-t001:** BioID2 constructs of various Ku70 segments. A simplified, short form construct name is denoted for each BioID2 fusion protein. NLS specifically refers to the Ku70 nuclear localization signal. Note that the Ku70 vWA also includes the addition of a Ku70 NLS added to its C-terminus, before the biotin ligase.

Construct Name	Residues (aa)	Length (bp)	Fusion Protein Name	Size (kDa)
NLS	539–556	54	NLS-BioID2-HA	30
Ku70	1–609	1827	Ku70-BioID2-HA	96
Ku70 vWA	1–250	750	Ku70 vWA-NLS-BioID2-HA	60
Ku70 ∆vWA	251–609	1077	Ku70 ∆vWA-BioID2-HA	68

**Table 2 cells-10-00646-t002:** List of Ku70 ∆vWA-specific candidate protein interactors that appeared in three biological replicates of Ku70 ∆vWA-BioID2. Final Ku70 ∆vWA candidates are shared with full-length Ku70 results and absent from the Ku70 vWA protein candidates list. Final candidates were filtered using the negative control cell lines, HEK293 and NLS-BioID2.

Gene Name	Full Name
WRN ^1^	Werner syndrome ATP-dependent helicase
NKRF ^1^	NF-kappa-B-repressing factor
APLF ^1^	Aprataxin and PNK-like factor
ZFR	Zinc finger RNA-binding protein
RPL35A	60S ribosomal protein L35a
HSPE1	10 kDa heat shock protein
ASPM	Abnormal spindle-like microcephaly associated protein

^1^ Known Ku interactors.

**Table 3 cells-10-00646-t003:** List of Ku70 vWA-specific candidate protein interactors that appeared in three biological replicates of Ku70 vWA-BioID2. Final Ku70 vWA candidates are shared with full-length Ku70 results and absent from the Ku70 ∆vWA protein candidates list. Final candidates were filtered using the negative control cell lines, HEK293 and NLS-BioID2.

Gene Name	Full Name
YLPM1	YLP motif-containing protein 1
WAC	WW domain-containing adapter protein with coiled-coil
FAM192A	Protein FAM192A
NFATC2IP	NFATC2-interacting protein
ZNF830	Zinc finger protein 830
ALKBH5	RNA demethylase ALKBH5
API5	Apoptosis inhibitor 5
SPDL1	Protein Spindly
RFX5	DNA-binding protein RFX5
RNF113A	RING finger protein 113A
SHOX2	Short stature homeobox protein 2

## Data Availability

The mass spectrometry data detailed in this manuscript has been submitted to the ProteomeXchange Consortium via the PRIDE partner repository with the dataset identifier PXD020165 and 10.6019/PXD020165.

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
