# Peer review of "Identification of Ku70 Domain-Specific Interactors Using BioID2"

_cells, 2021, doi:10.3390/cells10030646_

Round 1

Reviewer 1 Report

In the manuscript titled “Identification of Ku70 domain-specific interactors using BioID2”, the authors  generate candidate lists of Ku70 domain-specific interactors by using BioID2 proximity labeling and then validate a few selected interactions, namely RNF113A and Spindly by proximity ligation assays.

The two major findings of the paper can be summarized as follows:

  • The BioID2 technique is suitable to find interactors that are specific to the vWA domain of Ku70, but suffers from both false positive and false negative hits

  • Both RNF113A and Spindly co-localize with Ku70 under physiological conditions without DNA damage in a vWA domain-specific manner

The design of this study is very well thought out. By creating four cell lines stably expressing the constructs NLS-BirA*, Ku70-BirA*-HA, vWa-NLS-BirA*-HA or ΔvWA-BirA*-HA, it was possible to identify interaction candidates either exclusive to the vWA domain of Ku70 or to the residual Ku70 protein. Considering the crucial role of the heterodimer Ku70/Ku80 for DNA repair via non-homologous end-joining, it is important to determine and characterize the difference in Ku70 interactors under both physiological conditions without DNA damage or under conditions of DNA damage. This manuscript provides a very nice breakdown of how to set up a screening for potential interactors of a specific protein domain with BioID2. Importantly, the drawbacks of this approach are also discussed, giving the reader a complete picture. The provided data are of high quality and convincing. However, there are some minor issues that need to be addressed to solidify the results and make some of the arguments clearer:

Minor:

  • Figure 2c: A quantification of the immunofluorescence microscopy images is missing. The mean nuclear intensity of the different constructs would give better insight into how homogenous the expression levels are in different cells of the same cell line, but also between cell lines.
  • Figure 3a: Line 464: “Ku70 domain constructs are smaller and expressed at higher levels, which could be contributing to the increased number of interactors identified (see Figure 3a)” While this explanation makes a lot of sense, the biotinylation level seems to be similar in the Streptavidin Western Blot (Figure 3a).
  • Figure 4: Regarding the MS data, what criteria were chosen for a protein to be identified (number of unique peptides and/or protein coverage? These additional parameters of the MS data would be useful for the reader to judge the quality of the proteomic data and conclusions of the study.
  • Figure 6: I may be wrong, but it seems like the contrast is not the same in all microscopy images. This makes it difficult to compare individual PLA signals. Also, a quantification of the number of PLA foci would be very helpful to underline the conclusions. Furthermore, at least some PLA conditions show a large number of foci in the nucleus, which makes it difficult to see individual foci. Some close-up visualization of individual nuclei could be useful here. Finally yet importantly, a control with a single primary antibody (e.g. anti-Ku70) would be more convincing than the control with both primary antibodies missing at the same time.

  • Line 71-72: I am aware of at least two studies, in which non-full length proteins were fused to BirA*. Here, the proxisomes of different MYC homology box deletions are generated: https://www.sciencedirect.com/science/article/pii/S1097276518308025?via%3Dihub

Here, the proxisomes of different GFI1B deletions were generated:

https://mcb.asm.org/content/39/13/e00020-19-0

However, this does not mean the author’s approach is not novel. In contrast to the two cited studies above, this manuscript provides clearer and more logical filtering of potential domain-specific interaction on candidates.

  • Line 502: Another reason why there are known Ku70 interactors in the smaller construct proxisomes, but not in the full-length Ku70 proxisome could be a stronger sterical hindrance of the BirA-tag on the full-length Ku70 than on the shorter constructs. Also, in the case of TERF2IP it could be that there just are not many accessible lysine residues for efficient biotinylation. The authors should also include these points in their discussion besides the points they mention, such as low abundance of the interaction partners or tagging the suboptimal terminus of Ku70.

Reviewer 2 Report

This manuscript by Sanna Abbasi and Caroline Schild-Poulter used a variation of the BioID approach to identify domain-specific protein interactions for Ku70. This is a continuation of a previous study based on BioID of Ku70 performed in their lab, and this new approach allowed the identification of additional/different interactors. They thus performed BioID2 (or BirA*?) with the Ku70- vWA domain deleted, or with that domain alone.

I believe the study is interesting and could identify novel interactors of Ku70. However, I have a concern regarding the constructs used. They authors are describing the different variants of BioID that have been developed over the years, and are mentioning in the abstract, in the introduction and in the materials and methods that they cloned and are using BioID2 in this study. However, the results section often mentions BirA* as the enzyme for this study. Figure 2a shows BirA*, while Figure 2c shows BioID2. BirA* is the E.coli enzyme which is found in the first version of BioID, and not BioID2. The authors need to verify and correct the text according to which enzyme was used in this study. The switch between BirA* and BioID2 appears throughout the text and within the figures.

While I understand the filters that were used, I think there are better approaches to filter rather than simply using the presence or absence of identification of a protein, such as either MS/MS counts or intensities. This can be achieve using SAINT for example, and filter based on scores. Perhaps a protein such as MRE11A which was filtered out because it was identified in one replicate of the negative control, yet maybe it is still highly enriched looking at other variables. The same is possible when comparing the appearance of MRE11A in Ku70 vWA and Ku70 deltavWA. Perhaps the protein in highly enriched in one of those experiment, yet this is missed based solely on the identification without quantification. I think the data presented would be greatly improved with a proper analysis. I think the fact that none of the vWA interactors could be confirmed by co-IP strengthen the need to use a different approach to identify interactors.

Additional comments:

  • The introduction mentions several advantages of BioID over other methods such as co-IP, but I think it would be interesting to underline the limitations as well.
  • The concept of using protein domains, or reconstructing different regions, coupled to AP-MS has been described before (fragmentome) and should be cited accordingly, not just yeast two-hybrid.
  • It is surprising to not see Ku80 in Table 2?
  • In Figure 6a, why was the full length Ku70 not used to validate the interactions?

Round 2

Reviewer 2 Report

While I still have some concerns regarding how the data was analyzed, particularly considering that hits that were only present in 1 of the three repeats were actually considered, I think the paper is now acceptable. It would have been further improved by applying more stringent parameters which would have resulted in less proteins, but more overlap in Figure 4 and 5.